# Demonstration of the Holonomically Controlled Non-Abelian Geometric Phase in a Three-Qubit System of a Nitrogen Vacancy Center

**DOI:** 10.3390/e24111593

**Published:** 2022-11-02

**Authors:** Shaman Bhattacharyya, Somnath Bhattacharyya

**Affiliations:** Nano-Scale Transport Physics Laboratory, School of Physics, University of the Witwatersrand, Johannesburg, 1 Jan Smuts Avenue, Wits 2050, South Africa

**Keywords:** qubits, NV center, holonomic control, non-abelian geometric phase, spin-orbit interactions

## Abstract

The holonomic approach to controlling (nitrogen-vacancy) NV-center qubits provides an elegant way of theoretically devising universal quantum gates that operate on qubits via calculable microwave pulses. There is, however, a lack of simulated results from the theory of holonomic control of quantum registers with more than two qubits describing the transition between the dark states. Considering this, we have been experimenting with the IBM Quantum Experience technology to determine the capabilities of simulating holonomic control of NV-centers for three qubits describing an eight-level system that produces a non-Abelian geometric phase. The tunability of the geometric phase via the detuning frequency is demonstrated through the high fidelity (~85%) of three-qubit off-resonant holonomic gates over the on-resonant ones. The transition between the dark states shows the alignment of the gate’s dark state with the qubit’s initial state hence decoherence of the multi-qubit system is well-controlled through a π/3 rotation.

## 1. Introduction

The study of the phase has produced a large number of applications particularly, in holonomic quantum computation by manipulating the circuit parameters where the gates applied on qubits are based on geometric phases [1,2,3,4,5,6,7]. The geometric (or Berry) phase is acquired by the adiabatic evolution of a quantum system around a circuit (path of rotation) which can be Abelian if the energy levels of a system are non-degenerate [2,5]. In our work adiabatic evolution implies that the qubit rotation causes a small change in the rotation angles. Conversely, if a system’s energy levels are degenerate, the cyclic evolutions of the degenerate subspaces will produce a path-dependent transformation called the non-Abelian phase where the angle or rotation is large [6]. The supposition that non-Abelian geometric phases can be applied to holonomic control of qubits was postulated over two decades ago [1]. Since then, holonomic control of qubits has progressed from speculation to the realization of control of physical qubits using their non-Abelian geometric phases [2,3,4,5,6,7,8,9]. Such geometric phases have been implemented by a single cycle of non-adiabatic evolution which is achieved by rotating a single qubit with a holonomic gate [10,11,12].

In a qubit, this geometric phase will have a non-zero solid angle that can be used to store information about the acquired geometric phase. The trajectory of the unit vector along the Bloch sphere is equal to half of the solid angle. The pure geometric phase arises from the geometric property of the closed loop or path that can be extremely resilient to the effects of noise since noise does not cause any changes to the area enclosed by the loop. Since the Berry phase is only dependent on the global geometry of the loop and is resistant to small errors, quantum gates that rely on the geometric phase have higher performance than other types of gates. Such quantum gates can be protected against the effects of decoherence, which may improve the performance of such gates. Further progression is seen in the development of multi-qubit holonomic gates, both in superconducting qubits and ion traps [6,7,8].

Nitrogen vacancy (NV) centers in diamonds show great promise for holonomic control due to having both a stable non-Abelian geometric phase and the means of control via optical excitation [10,11,12,13]. Qubits in NV centers can be rotated by microwave, radiofrequency, and Stokes laser pulses [12,13,14,15,16,17]. These pulses can be used to implement a set of universal quantum gates which then operate on the qubits. Such control of NV centers via non-Abelian geometric phases has been demonstrated, with holonomic single-qubit gates implemented on NV centers demonstrating high fidelity, even at room temperature [2,11]. Recently, the use of polarised microwaves on NV centers has demonstrated the ability to perform universal quantum gate-based operations on multiple qubits [9]. Through the creation of two separate cycles on two separate qubits which are then concatenated, a geometric phase will be created [18,19]. However, there are very few attempts to show the holonomic control of three coupled qubits [20,21,22,23]. In particular, a method of performing holonomic control of three-qubit systems (three three-level Rydberg atoms) using a single holonomic gate is proposed; however, NV centers have specific energy levels of transition [22]. In our previous report, we demonstrated a quantum simulation of two levels of an NV center by an IBM QE [24]. Recently, we also showed three qubits connected by three coupling parameters without considering any geometric phase [25]. In this work, we demonstrate the holonomic control of three coupled qubits in addition to the replication of some well-known results which provide a solid foundation for simulating holonomic control of NV-centers using IBM QE (see References [10,11] and Appendix A).

A three-qubit system produces a similar effect to Rashba spin–orbit coupling (RSOC) and vortex structures in materials which are highlighted in the paper [26,27,28]. There have been attempts to acquire the geometric phase in a qubit circuit of IBM QE [21,29,30,31]. *P*_r_ shown in Figure 1b–e is defined as the probability of the desired state (most often, the initial state) after measurement. The improved probability of electrons in the asymmetric arrangements of three entangled qubits has been demonstrated [28]. However, a set of three qubits accumulating a geometric phase dependent on time has not been achieved [32,33,34,35]. A holonomic gate can be defined as an operation or gate which causes a qubit rotation such that the qubit acquires an Abelian phase which can be implemented on qubits in optical systems such as NV centers or superconductors [36,37,38,39,40]. For the three-qubit system in a trigonal arrangement that occupies a closed space, we can generate a non-Abelian geometric phase and show the superiority of off-resonant holonomic gates over the on-resonant gates. This development shows particularly interesting potential as NV centers are viable qubits at room temperature, and the operation of universal gates on such qubits is a step forward in the development of universal quantum computers.

Another very important property of a holonomic gate in an NV center is that it also has a dark and bright state. This property and the initialized state of the qubit play a role in the gate’s performance by controlling the level of decoherence in the system. According to Reference 10, the decoherence for a single qubit system reaches a minimum when the dark states of the gate align with the initial state of the qubit. For 2 qubits or a 4-level system, |D_1_> and |D_2_> were demonstrated [11]. Here, we show the transition between three dark states in a three-qubit system.

## 2. Results: Holonomic Control of Qubits

### 2.1. One Qubit

To implement holonomic control on multiple qubits, we first demonstrate the operation of a holonomic gate on a single qubit. The circuit for the implementation of a one-qubit holonomic gate consists of three main processes, namely, the initialization of the qubit, the holonomic gate implementation, and the measurement (Figure 1a). We start by initializing a Bloch sphere to the states |z〉 and |−z〉 since the final states of the qubit are measured on the standard $\left|z\right.$ basis. We also initialize the standard states |x〉 and |y〉. Unlike an ordinary qubit where |z〉 =0 and |−z⟩ = 1, in this Bloch sphere the state $\left|z\right.=|1>$ and the state |−z⟩ = |2>. Here, |1> and |2> represent two non-degenerate ground states of the NV center. A set of pulses is used to implement a single qubit unitary holonomic gate U with parameters *θ*, *ϕ*, and *λ*. Here θ and *ϕ* are used to denote the rotation angles in the Bloch sphere and λ=Δ/Ω where Δ and Ω correspond to the detuning frequency of the pulse and Rabi frequency of the qubit, respectively. The holonomic gate on IBM QE θ and *ϕ* is implemented using R_X_, R_Y_, and R_Z_ gates, respectively. The pulse with the frequency Δ is appended into the circuit to achieve the detuning. To this end, we reproduce some of the previous work through emulations on IBM QE [2,10].

We start by characterizing a simple rotation path that only has one axis of rotation such as a π/2 rotation around the Y axis (Figure 1a). Now, we continue by characterizing arbitrary rotations around the x-axis via an R_X_ gate and other gates of the set X(*γ*) with *ϕ* = 0 where the phase shift *γ* changes as a function of the detuning frequency (Figure 1b). This includes the X gate which corresponds to γ=π. This is extracted from the difference between the input and output states and expanded to show the holonomic rotations around arbitrary axes. We keep *ϕ* constant while θ is the variable. The discrepancy between the emulated and simulated transfer increases with θ starting from 0 and attains the largest values when θ=π as the initialized state aligns with the dark state of the qubit. It can be concluded that the difference between the emulated and simulated result increases as θ increases, but the discrepancy can be reduced if the initial state aligns with the dark state of the holonomic gate. The rotation of the qubit with variable detuning frequency is shown in Figure 1b,c by measuring the *P*_r_ on the |1〉 and |2〉 basis, with the most observable phase change seen when the qubit is initialized to |1〉.

By changing the axis of rotation and by increasing angle θ, keeping *ϕ* = 0 constant, we demonstrate effective Rabi oscillations between |1〉 and |2〉 as shown in Figure 1b). Similarly, by keeping θ=0 constant and changing *ϕ* Rabi oscillations can be demonstrated [14], implying holonomic control over the (*θ*, *ϕ*) degrees of freedom (Figure 1b,c). We also demonstrate tuneable rotations of an NV center around the x-axis and y-axis using a variable detuning frequency (Δ) that produces a phase shift in the qubit. The detuning frequency is varied over a range of frequencies that span resonance, resulting in the Rabi oscillations shown in Figure 1c. Since the holonomic gate is represented by a unitary matrix which is part of the SU(2) group, the actual phase accumulated cannot be directly measured. The discrepancy between the emulated and simulated results can be used to calculate the accumulated phase. However, when we keep detuning the frequency of a variable and hold *θ* at π/2, we realize that the discrepancy is the greatest at Δ = 0 and varies as a function of Δ. The measured fidelities of the X(π/2) and Y(π/2) are 0.84 and 0.86, respectively.

To determine the potential performance of the single-qubit holonomic gate in IBM QE, we also use sequences of three gates to achieve holonomic rotation as shown in Figure 1d,e. Using the same techniques for holonomic control already demonstrated, we emulate two further single-qubit rotations in the |z〉 basis. The *P*_r_ are shown in Figure 1d,e. The first rotation intersects the states |1〉 and the $1/\sqrt{2}\;\left(\left|1\rangle +i\right|2\rangle \right)$ on the surface of the Bloch sphere. The second rotation traverses the circumference of the Bloch sphere about an axis defined by an angle θ=π/4. The states do not cross because the maximum value of *θ* is less than π/2. It was rotated such that the *P*_r_ of state |1> reaches a minimum of greater than 50%. The path of rotation is like that of the second diagram of Figure 1a. After initializing to state |1〉, the gate transfers the population from |1〉 to |2〉 around the x-axis and back to |1〉 as the frequency of the pulse is altered. For the composite holonomic control, we observe that the maximum discrepancy between the emulated and simulated results is lower than that of the single holonomic gate control. This shows that the accumulated phase from multiple rotation gates is smaller than that of the single holonomic gate. Moreover, the fidelities of the composite gates reach an average of 0.77 which is lower than the single gate. It can also be concluded that using a single loop scheme is generally more efficient than using a complex gate that concatenates separate cycles because the fidelity of the single loop scheme (of around 0.87) is higher than the fidelity of 0.66 of complex gates. The fidelity of the single qubit gate is low due to the high noise levels of the chosen device. However, the use of multiple holonomic gates on one qubit has a similar efficiency compared to a single unitary gate. This analysis supports the results shown previously [10].

### 2.2. Two Qubits

For **two qubits**, the holonomic control is demonstrated in a four-level system [9,11]. We have considered oscillations between two ground states (|1〉=|00> and |2〉=|01>) and two excited states (|3〉 =|10> and |4〉=|11>) in a negatively charged NV-center. Using these approximations as the parameters of an R_x_ gate on each qubit of a two-qubit circuit results in the required π/2 gate being performed on the qubits (Figure 2a). The holonomic control of a 4-level system has been demonstrated in Figure 2b. When attempting to simulate this on IBM QE, the envelope and wavefunction had to be programmed (see Appendix A) [11]. Figure 2c shows the evolution of the state |00> for two qubits having combined adiabatic and non-adiabatic rotations.

### 2.3. Three Qubits

In the three-qubit holonomic control procedure, we use the Greenberger–Horne-Zeilinger (GHZ) frame of reference when mentioning any qubit states. The GHZ state is an entangled quantum state which is widely used for a three-qubit system [41]. It is difficult to demonstrate all the 9 levels of an NV center using qubits because for N number of qubits the total number of levels is 2^N^. Therefore, most holonomic simulations omit the |0> state and replace the poles of the first qubit with the states |1> and |2> (this corresponds with most holonomic simulations). The system in the rotating frame is given as:(1)$$H=\sum_{i=1}^{8}|\omega_{i}|i><i|+\frac{i}{2}(\Omega_{P_{1}}e^{-iV_{P_{1}}t}-\Omega_{P_{2}}e^{-iV_{P_{2}}t})\;(\left|1><7\right|-\left|1><8\right|)-\frac{i}{2}(\Omega_{S_{1}}e^{-iV_{S_{1}}t}\;–\Omega_{S_{2}}e^{-iV_{S_{2}}t})\;(\left|2><7\right|+\left|2><8\right|)+\frac{i}{2}\left(\Omega_{P_{3}}e^{-iV_{P_{3}}t}\right)\left(\left|1><3\right|\right)\;-\frac{i}{2}(\Omega_{S_{3}}e^{-iV_{S_{3}}t})\;\left(\left|2><4\right|\right)$$

The energy diagram for a three-qubit system, the qubit operations, and the desnsity of states are shown in Figure 3a–c. Δ_1_, Δ_2_, and Δ_3_ are one photon detunings between the various energy levels of the NV center. Δ_1_ is the detuning between the ground states |1> and |2> and the excited states |5> and |6> since |1> and |2> are degenerate ground states and |5> and |6> are degenerate excited states. Δ_2_ is the detuning between the ground states and the excited states |7>. Δ_3_ is the detuning between the ground states and the excited states |8>. The paths are concatenated, and the system is measured on the z basis to obtain the *P*_r_ of the state |1>. The order of measurement is important. The first qubit is measured first; the second qubit is measured after the first qubit, and the third qubit is measured last.

In the first scenario (Figure 4a), we only show the holonomic control of the qubits around the standard x, y, and z axes. We use an R_X_ gate to rotate the first qubit by *θ* = 5π/6. We do the same for the second qubit; however, the parameter for this R_X_ gate is −π/6. The third qubit is rotated by π/3 (*θ* = 2π/3). These steps above are used to initialize the system. In a similar manner to the two-qubit scenario above, we consider three rotation paths. For the first qubit, we implement a holonomic gate with *ϕ* = 0 and *θ* = 2π which is a complete rotation around the y-axis. The Rabi frequency for the first qubit Ω_1_ is equal to the qubit frequency stated in IBM QE at 4966 MHz. A similar holonomic gate was implemented on the second qubit. For this holonomic gate *θ* = −2π which is a complete rotation in the opposite direction from the first qubit. The third qubit is rotated around a circular path around the y-axis with *θ* ranging between π/6 and −π/6. The same is true for *ϕ*. We keep the detuning frequency for the first holonomic gate Δ_1_ variable for varying values of the detuning frequencies for the other holonomic gates Δ_2_ and Δ_3_.

The paths are concatenated, and the system is measured on the z basis to obtain the *P*_r_ of the state |1>. In Figure 4a, a simulation shown by the solid red line represents the ideal scenario where Δ_1_ = Δ_2_ = Δ_3_. The probabilities of the |1> state start at 100 and decrease to 0 as the rotation angle is π. We also notice that there is a discrepancy between the emulated and simulated results. As mentioned for the one (and two-qubit) scenario, this discrepancy indicates that an effective phase is accumulated across all three paths. The nature of this phase is Abelian for each path. However, the combined effect of all the paths produces a non-Abelian geometric phase because both d*θ* and d*ϕ* are non-zero. The magnitude of the phase varies as a function of the detuning frequency. The magnitude reaches a maximum value at Δ_1_ = 300 MHz. The phase increases with the difference between the values of Δ_2_ and Δ_3_. This is consistent with the fact that off-resonant holonomic gates have a higher performance compared to on-resonant gates which do not allow significant phase accumulation.

In the second scenario (Figure 4b), we consider a similar configuration for the system; however, we measure the effective rotation time for the varying values of Δ_1_, Δ_2_, and Δ_3_. We measure the rotation time of each of these paths as well as the effective rotation path to obtain the *P*_r_ of the state |1>. Each rotation accumulates a geometric phase which is Abelian so that d*ϕ* = 0 for the first two rotation paths, and only either d*ϕ* or d*θ* is non-zero for the whole rotation. Now, since calculating the exact phase matrix requires the evaluation of path integrals and the calculation of gauge potentials, we approximate the effective phase matrix:(2)$$U_{eff}=\left(\begin{array}{cc} 0.99+0.47i & -0.82+0.12i\\ 0.93+0.82i & 0.65+0.33i \end{array}\right)$$

Although each separate rotation represents Abelian geometric phases, the effective phase can, however, be realized non-Abelian. This explains why *U*_eff_ is not unitary. Our results are consistent with Reference [2]. Some details of the operation for a two-qubit system are presented in the Appendix A.

## 3. Discussions

### 3.1. Dark States

In qubits, a dark state is a state where a qubit undergoes trivial dynamics (i.e., the qubit can be described by the qubit Hamiltonian). For the creation of geometric phases, the rotational path traversed by the qubit is usually a closed loop and can be implemented on an arbitrary axis such as the one shown in Figure 1a. The evolution of dark states for a two-qubit system is given in the Appendix A (see Appendix A). In Figure 4a, it can be observed that the initial return probability at zero Δ_1_ for the dark states is the same for all values of Δ_2_ and Δ_3_. Moreover, the deviation from the classical approximation is the least at zero Δ_1_. This indicates that the dark state of the gate is aligned with the initial states of the qubits. This is consistent with the results obtained for a single qubit in Reference 10. It is interesting to note that with the increase of the detuning frequency, more deviations occur in the return probability. Under normal circumstances, this deviation is mostly related to decoherence in the systems. However, due to the nature of the qubit rotation, this deviation is also related to the creation of a geometric phase. This geometric phase is observed to be partially dependent on the detuning frequency. Moreover, the gate’s dark state seems to align with the initial state of the qubit at 450 MHz as the deviations decrease to a minimum.

In Figure 4b, a similar conclusion can be drawn regarding the alignment of the dark state of the holonomic gate and the initial state of the qubits. At *t* = 0, the dark state of the gate aligns with the initial state of the qubit. This can be seen by the fact that at *t* = 0, the return probability is the same as the *P*_r_ of the simulated results. Both Figure 4a,b indicate that the alignment of the dark and initial states allows for the control of decoherence in a system consisting of multi-qubits. Figure 4c shows the evolution of the states of the system caused by rotating the second qubit in the system. Only one qubit is rotated so that this rotation can be described by the quantum Rabi model and subsequently a Rabi envelope can be developed. The amplitude of the state |5> increases to 0.86 while the state |1> drops to 0.86. This shows that a π/3 rotation has been achieved. The excited states do not have a non-zero population once the π/3 rotation is complete. Moreover, it is seen that the other states follow the same trend. The increase in amplitude for those states is consistent with the transition between the three dark states of our qubit system. To prove the presence of a dark state in a system, the eigenstates of the interaction Hamiltonian using Equation (1) are calculated as shown in Figure 4d, which is consistent with Figure 4c. The three-qubit off-resonant holonomic gates show higher fidelity (~80%) compared to the on-resonant gates (~70% fidelity).

Dark states can be created by the strong overlapping of two orthogonal states like the spin-orbit interaction. In condensed matter physics, it is well known that the geometric phase arises from the RSOC. This, however, has not been experimentally demonstrated using a quantum simulator that requires at least three qubits. Earlier, a three-qubit system was employed to generate a synthetic magnetic field, a chiral ground state current, or a chiral spin state; however, the geometric phase was not captured which was attempted in the present work [29]. The purpose of the holonomic operation is to create a geometric phase through a closed loop, through multiple iterations. Dark states can be described as the bound states or the vortex core which arises from the strong spin–orbit coupling which results in weak (anti-) localization phenomena without breaking the time-reversal symmetry as observed in Figure 4b [21]. This picture becomes very interesting for three vortices that can be controlled by a single operation. We see fine control of the *P*_r_ with the Rabi frequency (or RSOC strength). At the zero frequency, *P*_r_ has a peak at the origin or zero detune time (like the weak localization phenomena observed in the magnetic field-dependent resistance of metal with the effect of RSOC). This becomes a minimum by the application of the Rabi frequency which is like weak anti-localization [31]. This is a hallmark feature of RSOC in a topologically protected material. While two holonomic operations perform a weak localization-like feature, the third holonomic gate breaks the time-reversal symmetry. Hence the operation of a three-qubit system works more efficiently than the two-qubit system [11].

The one qubit system can only be encoded with the ground states of the NV centers (or with one dark and one bright state). This means that only one dark state can be measured. However, since the ground states are degenerate, no transitions are possible. For a two-qubit system, we observed two dark states. The state tomography for the two-qubit systems presented in Figure 2d shows the rotation path for the first qubit as a circular path. The second path is a square loop with each side of the square created by rotations about the x and z axes. The rotation of the second qubit is dependent on the rotation path of the first qubit for transitions between the first and second dark state to occur as seen in Figure 2d.

For the three-qubit system, since there are more dark states, transitions between the dark states can be observed just through the holonomic control of one of the three qubits in the system. The transition is also quite interesting because it causes the amplitudes of the states other than the initial states to increase for a short duration. Moreover, with three-qubit systems, the system can exhibit geometric dependence which is not possible for one- and two-qubit systems since only one configuration exists for these systems while there are two configurations for the qubits in the three-qubit system. Quantum state tomography for the three-qubit system is similar to the two-qubit system. We see in Figure 3b,c that the rotation of the third qubit is dependent on the motion of the second qubit, which in turn is dependent on the first qubit. This allows for a transition to occur between the first, second, and third dark states of the NV center implemented by the three qubits.

### 3.2. Fidelity

Finally, the fidelity of the states is calculated using the formula:(3)$$F\left(\rho ,\kappa \right)=Tr{\left[\sqrt{\sqrt{\rho }\kappa \sqrt{\rho }}\right]}^{2}$$

Here, *ρ* represents the density matrix of the ideal state (simulated using the QasmSimulator of IBM). The density matrix of the state represented by *κ* is obtained by performing a digital simulation using an actual IBM quantum computer. We use the IBM Q 14 Melbourne for all the digital simulations. We perform the calculation of the fidelities of both the on- and off-resonant gates separately. The variation of the fidelities for both the on- and off-resonant gates in Figure 3c are shown as functions of time.

It can be seen from Figure 5 that the fidelities for both the off-resonant and on-resonant are initially low at about 60%. This is because the IBM Q14 Melbourne device is quite noisy, and no error mitigation and noise control techniques were employed to reduce any decoherence in the system. According to IBM, the simulator noise model consists of depolarizing error followed by a single qubit thermal relaxation. However, as time increased, the fidelities of both gates also increased. Now, since this is holonomic control, the reason for this can be attributed to the creation of a geometric phase for both on- and off-resonant gates. However, the fidelity of the off-resonant gates increases to ~85%, which is significantly higher compared to that of on-resonant (~70% fidelity) gates. This shows that the performance of off-resonant gates is higher than that of on-resonant gates for the three qubits holonomic control.

## 4. Conclusions

IBM Quantum Experience is a powerful tool for simulating quantum mechanical systems such as NV-centers in diamonds, and the potential for simulating holonomic control of diamond-based spin qubits is very real. Before attempting the holonomic simulations of non-Abelian gates, the basic method of simulating NV centers has been demonstrated. Subsequently, the results of several different papers were reproduced using IBM QE simulations. We use an eight-level system and give a nearly complete description of an NV center. This can be compared to two four-level systems each corresponding to a vortex (a bound state) and their interactions. A set of three qubits also includes strong spin–orbit-like interactions, and we see all three dark states of the holonomic gates can be aligned with the initial state of the qubits, hence allowing control of the decoherence of the entire system through a π/3 rotation. We demonstrated the high fidelity (~85%) of three-qubit off-resonant holonomic gates over the on-resonant ones. For multiple qubits, the Dicke model is required which is beyond the scope of our work. This simulation can also be applied to understand the resonant properties of a three-quantum dot system [42]. Since this is the first attempt at simulating the three-qubit holonomic gate, details of the calculation of the exact nature of this system will be given elsewhere.

## Figures and Tables

**Figure 1 entropy-24-01593-f001:**
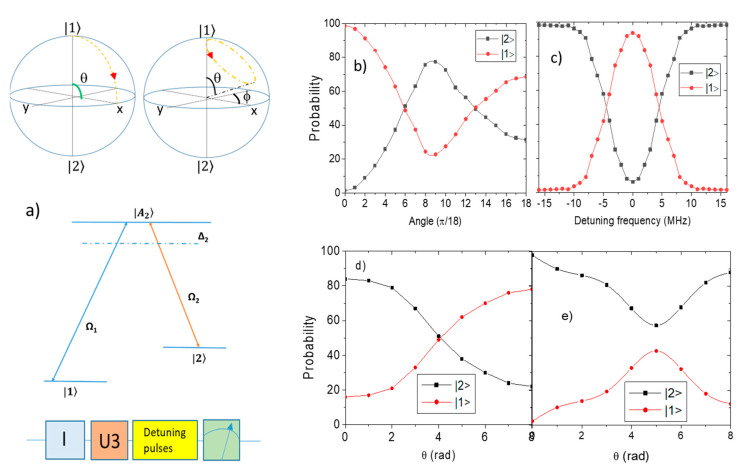
Simulation of holonomic control on a single qubit. (**a**) (top) The rotation path is implemented by a single holonomic gate. In the first diagram the qubit is rotated about the Y-axis by π/2. Since the path forms a closed-loop a geometric phase can be created via a single iteration. (middle) The NV center consists of a triplet ground state. The detuning is between the ground and excited state since these two states are the states that were used for the simulations. The detuning is between the exited state |A_2_> and the ground state |1> and |2>. Δ represents one photon detuning frequency. The axis is defined as n = (sin*θ*cos*ϕ*, sin*θ*sin*ϕ*, cos*θ*). By peaking arbitrary values of *θ* and *ϕ* the rotation axis can be chosen, e.g., if *θ* = π/2 and *ϕ* = 0 then the qubit is rotated by π/2 about the y-axis. (bottom). The circuit shown here is used to initialize the qubit and then implement the holonomic gate using a pulse schedule. Then, the system is measured on the |z〉 basis. (**b**) Gates with the angle θ variable after initializing the qubit to |1〉. The probabilities of the final states are to be measured on the |z〉 basis from both |1〉 and |2〉 and to be plotted against θ. The red and black lines indicate the behavior of the qubit in the presence of noise. (**c**) A holonomic gate with a variable detuning frequency and fixed rotation axis x is applied to the qubit. The oscillations for the qubit are shown which are caused when the detuning frequency causes the qubit to rotate around the x-axis. The red and black lines indicate emulations in the presence of decoherence. The phase change becomes more observable when the qubit is initialized to the state |1〉. (**d**) Emulation of multiple single qubit holonomic gates. Gates with the angle *θ* and *ϕ* variable where *θ* = *ϕ*. The qubit is initialized to 0 and the rotation axis is kept constant since *θ* = *ϕ*. The probability of the final states is obtained using the |z〉 basis for measurement. For the implementation of the rotation, we use an R_X_ and an R_Z_ gate. (**e**) The same set of holonomic gates is implemented on the qubit; however, the qubit is initialized by applying a π/4 rotation to the state |1〉. Red and black lines indicate emulations with decoherence. Both the phase and decoherence are reduced when the rotation axis of the qubit aligns with the dark state of the gates since it undergoes trivial dynamics.

**Figure 2 entropy-24-01593-f002:**
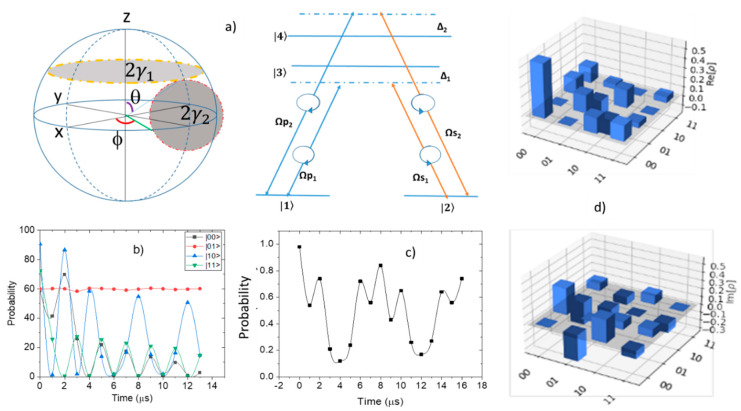
(**a**) (left) The two rotation paths of the qubits implemented in the NV center. These paths start and end at the same position on each qubit. The first path is implemented on the first qubit while the second path is implemented on the second qubit. (right) The effective energy diagram of the spin triplets (^3^A and ^3^E) for the 4-level NV center including the selection rule. The Rabi frequencies are Ω_pi_ for the applied pump and Ω_si_ for the Stokes field with *i* = 1 and 2. The detuning is Δ_1_ = Δ_2_ = Δ. (**b**) The return probability of the state |1> after evolution in the rotation path for varying degrees of the degeneracy of the ground states over time. (**c**) Evolution of |00> state over time for two qubits; one undergoes adiabatic and the other non-adiabatic rotation. (**d**) Quantum state tomography for the two-qubit systems showing real and imaginary parts of the density of states.

**Figure 3 entropy-24-01593-f003:**
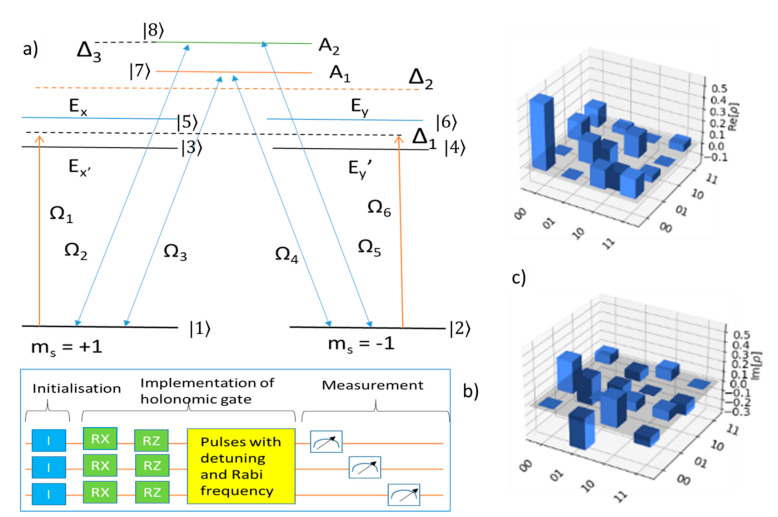
(**a**) Energy diagram of the spin triplets ^3^A and ^3^E for the complete 9 level NV center including the selection rule. The Rabi frequencies are Ω_pi_ for the applied pump and Ω_si_ for the Stokes field with *i* = 1, 2, and 3. The detuning is Δ_1_ = Δ_2_ =Δ_3_ = Δ. (**b**) Quantum circuit used for the holonomic control of three qubits. The identity gates (any other gate that can be used here) are used to initialize the qubits. R_X_ and R_Z_ gates are used to set the parameters θ and φ, respectively. The pulses are implemented with Δ_i_ and Ω on all three qubits to achieve full holonomic control. (**c**) Quantum state tomography for the three-qubit systems shows real and imaginary parts of the density of states.

**Figure 4 entropy-24-01593-f004:**
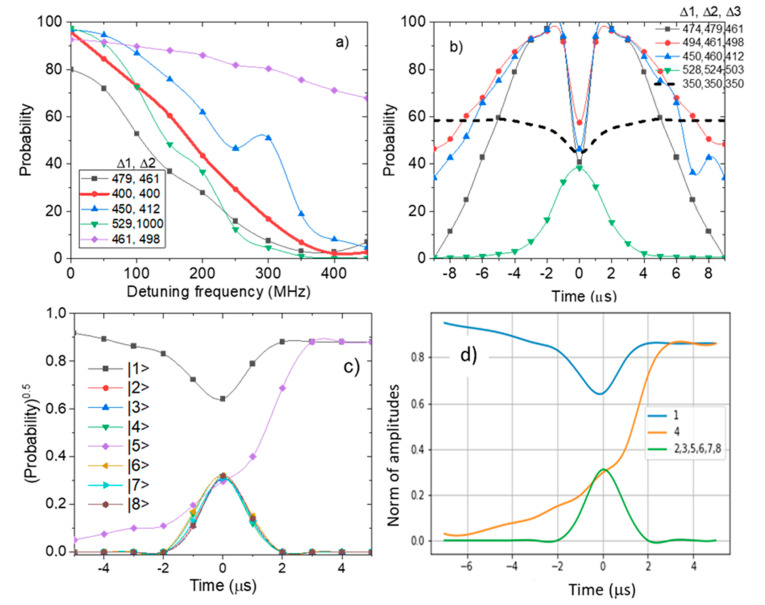
(**a**) Simulation of holonomic control on an eight-level NV center with detuning frequency variable for varying detuning frequencies of the three qubits. The rotation paths of each qubit are concatenated to form a single effective rotation path. The probabilities of the state |1> are plotted against detuning frequency Δ_1_ for fixed detuning Δ_2_ and Δ_3_. The solid line with open symbols represents the on-resonant case. (**b**) Evolution of the state |1> over pulse sequence duration for varying values of detuning frequencies Δ_1_, Δ_2_, and Δ_3_. The solid dashed line represents the on-resonant case. (**c**) The time evolution of the states over pulse duration illustrating a π/3 rotation. The grey square, purple diamond, and green triangles and lines represent the |1>, |5>, and |4> states, respectively. (**d**) Evolution of the state |1> as a function of time with varying values of detuning frequency as described by the interaction Hamiltonian in Equation (1).

**Figure 5 entropy-24-01593-f005:**
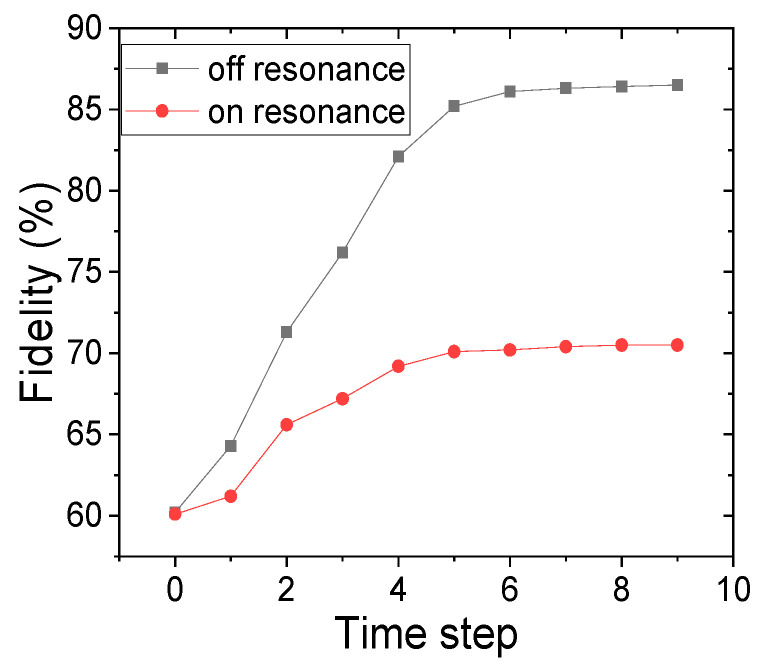
The change in the fidelity of the states obtained from the off-resonance and on-resonance gates as a function of time steps.

## Data Availability

The data that support the findings of this study are available from the corresponding authors upon reasonable request.

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
