# Peer review of "Demonstration of the Holonomically Controlled Non-Abelian Geometric Phase in a Three-Qubit System of a Nitrogen Vacancy Center"

_entropy, 2022, doi:10.3390/e24111593_

Round 1

Reviewer 1 Report

The authors demonstrated the holonomically controlled non-Abelian geometric phase in a three-qubit system of an NV center. The geometric phase can protect quantum gates against the effects of decoherence, and the NV centers system is a strong competitor to realize quantum computation. This work has the potential application in the quantum computation. However, the manuscript cannot be recommended until the authors visit my following misgivings in the attaching pdf-file.

Author Response

First reviewer:

In this paper, the authors simulated holonomic control of non-Abelian geometric phase in an NV center system, based on IBM quantum experience technology. The results are consistent with their references, reflecting the evolution law of holonomic control under different parameter conditions, also including decoherence and noise environment. They also measured the density matrix of the quantum state using the quantum tomography technique. Overall, the results of this manuscript are reasonable and meaningful, while there are still some issues that need to be properly considered before publication. 

Reply: Dear Reviewer, many thanks for reviewing our paper and suggesting several valuable points to improve the presentation. Please find our point-by-point reply below. All points are included in the revised version of the manuscript.

  1. In Fig. 1.e), could the author explain why the curves of state |1⟩and state |2⟩ do not intersect?

Reply: The states do not cross because the maximum value of q is less than p/2. The qubit was rotated such that the probability of state |1> reaches a minimum of greater than 50 %. The path of rotation is like that of the second diagram of Fig 1a. This discussion is included in the revised version of the manuscript on page 4.

  1. What is the difference between adiabatic rotation and non-adiabatic rotation?

Reply: For the purposes of our work adiabatic rotations refer to rotations where the change in the parameters q and f are small. Adiabatic rotations can be thought of as infinitesimal rotations. Non-adiabatic rotations are rotations where the change in the parameters q and f are large. [added to the revised version in page 1).]

  1. In Fig. 3.a), what do Δ1, Δ2,and Δ3 mean, respectively?

Reply: The D1, D2 and D3 are one photon detunings between the various energy levels of the NV center.

D1 is the detuning between the ground states |1> and |2> and the excited states |5> and |6> since |1> and |2> are degenerate ground states and |5> and |6> are degenerate excited states. D2 is the detuning between the ground states and the excited states |7>.  D3 is the detuning between the ground states and the excited states |8>. This description is added in the revised version (page 6).

  1. In Fig. 3.b), is there any requirement for the measurement sequence of three qubits after the implementation of holonomic gate operation?

Reply: The order of measurement is important here. The first qubit is measured first, the second qubit is measured after the first qubit and the third qubit is measured last. This is added in the revised version (page 6, bottom).

  1. The caption of Fig. 4.c) seems to be wrong. Does the “green triangle” represent |4⟩state?

Reply: There was a mistake regarding the caption. This is corrected in the revised version (in figure caption 7).

  1. In Eq. 4, “Ueff” does not satisfy the unitary property. Why?

Reply: This is due to the non-abelian phase. A note is added in page 8.

  1. Which method does the authors use to achieve quantum state tomography?

Reply: We used IBM Qiskit which has a built-in program to perform a state tomography experiment. Density matrix of the final state is calculated.  The experiment is initialized with a circuit to prepare the state to be measured.

Please let us know should you need further improvement of the manuscript.

Thank you very much for providing us some of the most valuable comments to improve the paper.

Reviewer 2 Report

In this paper, the authors simulated holonomic control of non-Abelian geometric phase in an NV center system, based on IBM quantum experience technology. The results are consistent with their references, reflecting the evolution law of holonomic control under different parameter conditions, also including decoherence and noise environment. They also measured the density matrix of the quantum state using the quantum tomography technique. Overall, the results of this manuscript are reasonable and meaningful, while there are still some issues that need to be properly considered before publication. 

1.     In Fig. 1.e), could the author explain why the curves of state |1 and state |2 do not intersect?

2.     What is the difference between adiabatic rotation and non-adiabatic rotation?

3.     In Fig. 3.a), what do Δ1, Δ2, and Δ3 mean, respectively?

4.     In Fig. 3.b), is there any requirement for the measurement sequence of three qubits after the implementation of holonomic gate operation?

5.     The caption of Fig. 4.c) seems to be wrong. Does the “green triangle” represent |4 state?

6.     In Eq. 4, “Ueff” does not satisfy the unitary property. Why?

7.     Which method does the authors use to achieve quantum state tomography?

Author Response

Second reviewer:

The authors demonstrated the holonomically controlled non-Abelian geometric phase in a three-qubit system of an NV center. The geometric phase can protect quantum gates against the effects of decoherence, and the NV centers system is a strong competitor to realize quantum computation. This work has the potential application in the quantum computation. However, the manuscript cannot be recommended until the authors visit my following misgivings.

Reply: Dear Reviewer, many thanks for reviewing our paper and suggesting several valuable points to improve the presentation. Please find our point-by-point reply below. All points are included in the revised version of the manuscript.

2.1 One qubit

â‘  I cannot see the definition of “return probability”.

Reply: The electron return probability can be defined as the probability that a desired state is measured which can be misleading. In our work, the |0> state is the desired state, so the probability of the |0> state will be the return probability, however it can be confusing. Therefore, we have used ‘probability’ in the revised version (page 2). The figure levels are corrected accordingly.

② Please define before using the energy level of NV center like |1⟩.

Reply: |1> and |2> represent two non-degenerate ground states of the NV center which are added in page 3, top.

â‘¢ The noises are considered, but noise type and level are not indicated.

Reply: According to IBM simulator noise model consisting of depolarizing error followed by a single qubit thermal relaxation. (added to page 10).

â‘£ The parameters ?, ?, ?, and Δ are not indicated. The angle is the rotation angle about which axis and the detuning between which two energy levels.

Reply: q and f are shown in Fig. 1(a). The parameter d is removed since it has not been discussed in the manuscript.

D represents one photon detuning frequency.

The detuning is between the ground and excited state since these two states are the states that were used for the simulations.

The detuning is between the exited state |A2> and the ground state |1> and |2>.

The axis is defined as n = (sinqcosf, sinqsinf, cosq). By peaking arbitrary values of q and f the rotation axis can be chosen e.g., if q =p/2 and f =0 then the qubit is rotated by p/2 about the y-axis. The NV center consists of a triplet ground state with ms=1, 0, -1.

This has been explained in the revised version (page 2, figure caption).

⑤ The actual physical system coupling two qubits is not indicated.

Reply: The qubits in Fig. 1a) are not coupled. Two block spheres represent different rotation paths of one qubit. However, the diagram is improved in the revised version. A diagram showing the energy levels of an NV center is added (page 2, Fig. 1)

â‘¥ Please explain the trajectory in Fig.1 (a) and Fig. 2(a) in detail. ? is the half of the solid angle, but in Fig. 2(a), it cannot be seen.

Reply: In figure 1 (a) the trajectory is a p/2 rotation about the y-axis, the second trajectory is a rotation about an arbitrary axis. In figure 2(a) 2 rotation paths are considered. The first path is a complete azimuthal rotation at q = p/3. The second rotation is a closed loop which is formed by rotating the qubit by the y- axis.

The solid angle for figure 2 a) is shown in the revised version (page 5, Figure 2).

⑦ Some explanations for numerical simulation should be added.

Reply: We have shown numerical simulation for 3 qubits only in Fig. 4d) (not for one qubit) Eqn. 1 and 2 are used for the simulation, which is discussed in page 7). The simulations were performed on a classical simulator operated by IBM known as the IBM Qasm simulator. In each simulation the holonomic gates were replaced with equivalent non-holonomic gates and the probability of the states were measured. Since there is no noise involved nor a geometric phase was created in these simulations, the results of these simulations can be compared to the results of the emulations and any discrepancy between these results can be interpreted as the formation of a geometric phase. The simulations were performed since the geometric phase cannot be measured directly in the SU(2) group. For one qubit, these simulations are trivial since elementary gates have been used.

⑧ For the single qubit gate, the fidelity is very low. Please explain the reason.

Reply: The reason for the low fidelity is due to the noise levels of the quantum device used rather than due to the single qubit operation itself.

This explanation has been added in the revised version of the paper (page 4).

2.2 Two qubits

â‘  The specific energy level of NV centers needs to be explained.

Reply: This point has been explained in the supplementary information which is now added to the revised version of the paper A diagram showing the energy levels of an NV center is added (page 5, Fig. 2).

â‘¡ Some articles need to be added to explain the GHZ frame.

Reply: This is a very important point which has been explained in the revised version of the paper and two references (Ref. 41) are added (page 5, top).

2.3 Three qubits

â‘  A clear energy level diagram needs to be added at the beginning.

Reply: The diagram is improved by identifying the energy levels. (Page 6, figure 3).

â‘¡ The Basic vector of Eq. (4) should be indicated. There are many spelling mistakes in the article, please check.

Reply:

The Ueff in Eqn. 4 was obtained by approximating the path integral of each path as described in Ref. 2.

The basis vectors are |1> and |2> since the measurements are carried out in {|1>,|2>} basis (page 8).

This operation has been explained thoroughly in the Supplementary information as follows.

“We implement each path on each qubit. The starting point of the paths is created by applying an RX (π/3) rotation which is implemented by setting θ to -π/2 and φ to π/3 on both qubits. The first path is a complete azimuthal rotation around the Bloch sphere. The phase matrix is approximated as: ?= (−0.609+0.321?0.786+0.1?−0786+ 0.1?−0.609+0.321?). The second path is a complete loop defined by θ and φ. The closed loop is created by a complete rotation around the y-axis of the second qubit. The coordinates of this rotation given in terms of φ and θ are given by (θ, φ) = [(π/3, 0), (0, π/3), (-π/3, 0), (0, -π/3)]. Now both paths accumulate a geometric phase. Since the excited states are non-degenerate and only dφ or dθ is nonzero the resulting geometric phase is Abelian. We also interchange the paths to observe any difference in the geometric phase. Now a disadvantage of SU(2) and higher-order systems is that the resulting geometric phase cannot be directly observed. In IBM QE we attempt to show this geometric phase by measuring the qubits in the |z> basis to obtain the density of the ground states. We also determine the expected density of states in the absence of the geometric phase and decoherence.”

The manuscript has been checked thoroughly and grammatical errors are fixed.

Please let us know should you need further improvement of the manuscript.

Thank you very much for providing us some of the most valuable comments to improve the paper.

Round 2

Reviewer 2 Report

The authors have answered all the points I made. I remommend the manuscript to publish in Entropy.